# Halogenase-Targeted Genome Mining Leads to the Discovery of (±) Pestalachlorides A1a, A2a, and Their Atropisomers

**DOI:** 10.3390/antibiotics11101304

**Published:** 2022-09-25

**Authors:** Mengna Luo, Mengyuan Wang, Shanshan Chang, Ning He, Guangzhi Shan, Yunying Xie

**Affiliations:** CAMS Key Laboratory of Synthetic Biology for Drug Innovation, Institute of Medicinal Biotechnology, Chinese Academy of Medical Sciences & Peking Union Medical College, Tiantan Xili No.1, Beijing 100050, China

**Keywords:** pestalachlorides, halogenase, genome-mining, flavin-dependent halogenases (FDHs), atropisomers

## Abstract

Genome mining has become an important tool for discovering new natural products and identifying the cryptic biosynthesis gene clusters. Here, we utilized the flavin-dependent halogenase GedL as the probe in combination with characteristic halogen isotope patterns to mine new halogenated secondary metabolites from our in-house fungal database. As a result, two pairs of atropisomers, pestalachlorides A1a (**1a**)/A1b (**1b**) and A2a (**2a**)/A2b (**2b**), along with known compounds pestalachloride A (**3**) and SB87-H (**4**), were identified from *Pestalotiopsis* *rhododendri* LF-19-12. A plausible biosynthetic assembly line for pestalachlorides involving a putative free-standing phenol flavin-dependent halogenase was proposed based on bioinformatics analysis. Pestalachlorides exhibited antibacterial activity against sensitive and drug-resistant *S. aureus* and *E. faecium* with MIC values ranging from 4 *μ*g/mL to 32 *μ*g/mL. This study indicates that halogenase-targeted genome mining is an efficient strategy for discovering halogenated compounds and their corresponding halogenases.

## 1. Introduction

Halogenated compounds play a profound role in the pharmaceutical industry as halogen substituents can significantly impact the bioactivity and reactivity of organic compounds [1,2,3]. According to an economic report, 88% of the 100 top-selling drugs employed chlorine in their final pharmaceutical products or the manufacturing process [4]. Nature is an important source of halogenated compounds. To date, over 5000 halogenated natural products have been discovered from fungi, bacteria, algae, cyanobacteria, plants, et al. [5]. Amongst, fungi as the third kingdom in nature, contributed nearly one-fifth (988) of halogenated metabolites [6] and are expected to harbor many more halogenated natural products to be identified [7].

Nature usually orchestrates halogen-carbon bond formation by a variety of halogenases. Several types of halogenases have been identified so far, including heme- or vanadium-dependent haloperoxidases, *S*-adenosyl-L-methionine-dependent halogenases, nonheme-iron α-ketoglutarate-dependent halogenases, and flavin-dependent halogenases (FDHs) [2,8,9,10]. Amongst halogenases, FDHs are widely distributed across all kingdoms of life [11] and are particularly notable for their strong regioselectivity and substrate diversity [2,12]. Almost all FDHs have the following two conserved motifs: A flavin-binding motif GxGxxG, for binding of the diffusible flavin adenine dinucleotide (FAD) [3], and a structural motif WxWxIP, thought to prevent a monooxygenation reaction by blocking direct contact between the substrate and hydroperoxy flavin [13,14]. These signature motifs can be used as probes for promptly identifying putative FDHs from genomic sequences. [2,3]

Fungi are a rich source of flavin-dependent halogenases (FDHs). Up to now, twenty-three halogenases have been reported from fungi, twenty of which are FDHs [7]. As for the substrates, FDHs prefer electron-rich precursors, such as phenols, indoles, or pyrroles. The phenol-containing structure is the most common substrate of the identified fungal FDHs, such as GedL from *Aspergillus terreus* NIH2624, which dichlorinated the phenol unit of sulochrin to produce dihydrogeodin, PtaM from *Pestalotiopsis fici,* which assembled one chloride atom to isosulochrin in the pestheic acid biosynthesis, and GsfI from *Penicillium aethiopicum,* which decorated griseophenone C with one chloride atom. Apart from the identified FDHs, thousands of putative FDHs were inferred in fungal genomes according to bioinformatics analysis [7], which hints that there are more halogenated metabolites or haloganases in fungi to be awaiting exploration. With the development of bioinformatics and the increasing decrease in genome sequencing costs, genome mining has become a powerful strategy for discovering new natural products or unearthing cryptic biosynthesis gene clusters [15,16,17]. An increasing number of chemical scaffolds, such as unusual post-translationally modified ribosomal peptide linaridins [18], PKS-NRPS hybrid aspyridones [19], and noncanonical polyketide burkholderic acid [20], have been discovered by genome mining. Although genome mining often involves genetic manipulation, including heterologous expression, in vitro reconstitution, and activation of the BGC in the native host, the non-genetic method sometimes shows high efficiency for mining the metabolites with characteristic features that can be easily detected using specific analysis methods. Halogenated compounds often exhibit characteristic isotope patterns in their mass spectra due to the presence of chlorine or bromine atoms, which makes them readily detectable from a complicated background. Additionally, according to the Natural Products Atlas database [6], more than 99% of halogenated microbial natural products are chlorinated or brominated ones, which consolidates the power of LC-MS in genome mining of halogenated natural products.

As a part of our efforts to investigate new natural products [21,22,23,24,25], we used the fungal FDH GedL [26] as a probe to explore the halogenase-containing BGCs from our in-house fungal genome database. A putative halogenase gene, *ptlK*, was mined from an endolichenic fungus, *Pestalotiopsis rhododendri* LF-19-12, and further bioinformatics analysis disclosed that *ptlK* was located in a cryptic BGC *ptl*. Subsequently, LC-MS was employed to interrogate the production of halogenated metabolites. As a result, a family of potential chlorinated compounds with characteristic chlorine isotope patterns were detected in the crude extract of *Pestalotiopsis rhododendri* LF-19-12 culture in the M2 medium. LC-UV-MS guided isolation led to obtaining two pairs of atropisomers, pestalachlorides A1a (**1a**)/A1b (**1b**) and A2b (**2a**)/A2b (**2b**), along with known compounds pestalachloride A (**3**) [27] and SB87-H (**4**) [28] (Figure 1). Here, we reported their discovery, isolation, structural elucidation, and biosynthesis.

## 2. Results

### 2.1. Genome Mining of the Halogenase-Containing Biosynthesis Gene Cluster

Flavin-dependent halogenases (FDHs), the most characterized halogenases based on their substrates, can be categorized into the following five main classes: free-standing phenol, free-standing indole, carrier protein-dependent phenol, carrier protein-dependent pyrrole, and aliphatic FDHs [9]. GedL is a free-standing phenol FDH from *Aspergillus terreus* NIH2624 [26]. It is involved in the biosynthesis of geodin and halogenates the substrate at the late stage of biosynthesis [26]. Here, we used GedL as the probe to conduct tBlastp analysis on our in-house fungal genome sequences. An antiSMASH analysis was subsequently performed, and a gene *ptaK*, encoding a putative flavin-dependent halogenase with 51% amino acid sequence identity to GedL [26], was found to be contained in a cryptic BGC of endolichenic *Pestalotiopsis rhododendri* LF-19-12. Succeeding phylogenetic analysis showed that PtlK grouped with free-standing phenol FDHs (Figure 2), suggesting that its substrate might hold a phenol moiety.

Subsequently, LC-MS and OSMAC strategies were employed to exploit the production of halogenated secondary metabolites. *Pestalotiopsis rhododendri* LF-19-12 was cultured in four different media (M1, M2, PDB, and YES) and then extracted using MeOH. The obtained material was applied to LC-MS analysis. As a result, a group of potential halogenated compounds with characteristic isotope patterns of two chloride atoms were detected in the crude extract of the *Pestalotiopsis rhododendri* LF-19-12 culture in the M2 medium (Figure 3).

### 2.2. Structural Elucidation for (±) Pestalachlorides A1a, A1b, A2a, and A2b

*Pestalotiopsis rhododendri* LF-19-12 was fermented in the M2 medium, and pestalachlorides A1a (**1a**), A1b (**1b**), A2a (**2a**), A2b (**2b**), and A (**3**), as well as SB87-H (**4**), were isolated and purified by an LC-UV-MS-guided method from a 9-day broth culture. Briefly, the culture of *Pestalotiopsis rhododendri* LF-19-12 was filtered, and the obtained mycelia were extracted with acetone. The yielded crude extract was fractionated and separated sequentially using silica gel, an ODS flash column, and further purified by semi-preparative chromatography to yield **1a** (23.2 mg), **1b** (2.2 mg), **2a** (1.9 mg), and **2b** (0.2 mg).

The HR-ESIMS spectrum of compound **1a** revealed a characteristic isotope pattern of double chlorides (Figure 3). Furthermore, analysis of HR-ESIMS and ^13^C NMR data disclosed that **1a** has a molecular formula of C_23_H_26_Cl_2_NO_6_ ([M+H]^+^ *m/z* 482.1121, calcd. 482.1137). Interpretation of the ^1^H, ^13^C NMR, and HSQC data for **1a** (Table 1, Appendix A) disclosed a carbonyl group (*δ*_C_ 168.6), 12 aromatic carbons, one of which is protonated, a trisubstituted olefin, a methine, three methylene units, one of which is attached to an oxygen atom, four methyl moieties, one of which is methoxy, and three phenolic hydroxyl groups. All the above interpretations accounted for 8 degrees of unsaturation and required **1a** to incorporate three rings, two of which should be aryl rings.

^1^H-^1^H COSY correlations (Figure 4 and Appendix A) revealed two isolated proton spin-systems attributed to -CH_2_-CH_2_-OH and -CH_2_-CH= (Figure 4). Furtherly, an isoprenyl unit in **1a** was established by HMBC correlations (Figure 4 and Appendix A) from H-4’ and H-5’ to vinylic carbons C-3’ and C-2’. HMBC correlations from H-1’ and H-2’ to C-6 suggested that the isoprenyl group was connected to the aromatic ring at C-6. Two phenolic hydroxyl groups at C-5 and C-3, respectively, can be inferred by the downfield chemical shifts of C-3 and C-5. Further correlations from H-4 to C-2, C-6, C-3, C-5, and C-1, from H-8 to C-6, C-2, and C-1, as well as from H-16 to C-8 and C-1, allowed construction of the substituted isoindole-1-one scaffold.

HMBC correlations from H-1” to C-11, C-12, and C-13, from the phenolic proton at *δ* 10.03 to C-9, C-10, and C-11, from the methoxy protons at *δ* 3.05 to C-14, from H-8 to C-9, C-10, and C-14 indicated that a hexasubstituted benzene ring was attached to C-8 via a C-C bond. As a result, the two chlorine atoms in **1a** could only be located at C-11 and C-13. Therefore, the planar structure of **1a** was assembled as shown in Figure 4.

The structure of **1a** was further confirmed by the single crystal X-ray analysis. The crystallographic data disclosed that **1a** featured a centrosymmetric space group *P*121/*c*1, suggestive of its being a racemate of 8*R* and 8*S* enantiomers (Figure 5).

Compound **1b**, an isomer of **1a** by HR-ESIMS analysis, was quickly converted into **1a** in acetonitrile aqueous. Therefore, only a mixture of **1a** and **1b** was obtained. The ^1^H NMR spectrum of the mixture displayed the following two sets of signals (Appendix A): one set of signals is identical to that of **1a,** and the others are nearly similar to those of **1a** except for the methoxy proton chemical shifts (*δ* 3.97, deshielded for **1b** vs. *δ* 3.05, shielded for **1a**), suggestive of **1b** as an atropisomer of **1a**. The ^13^C NMR spectrum (Appendix A) further supported the above hypothesis. To our knowledge, the analog of **1a** and **1b**, pestalachloride A (**3**) from *Pestalotiopsis adusta*, also has atropisomer axial chirality due to the hindered rotation around the C8-C9 bond, but its two atropisomers could not be chromatographically separated [27].

The HR-ESIMS analysis of compound **2a** returned a molecular formula of C_25_H_29_Cl_2_NO_7_ ([M+H]^+^ *m/z* 524.1240, calcd. 524.1243). The ^1^H and ^13^C NMR signals of **2a** are very closely related to those of **1a** except for the signals of the *N*-substituent as follows: *δ*_C-16_ 38.4/*δ*_H-16_ 2.56, 3.63, *δ*_C-17_ 23.2/*δ*_H-17_ 1.69, *δ*_C-18_ 31.1/*δ*_H-18_ 2.16, and *δ*_C-19_ 174.0/*δ*_H-COOH_ 11.99, indicative of a fragment of -CH_2_-CH_2_-CH_2_-COOH (Appendix A). The above proposed *N*-substituent was further confirmed by ^1^H-^1^H COSY correlations of H-16/H-17/H-18 and HMBC cross signals from H-17 and H-18 to C-19 as well as from H-16 to C-1 and C-8, respectively (Appendix A). The upfield methoxyl proton signals at *δ* 3.04 indicated that the methoxy was located in the shielded area of the isoindole-1-one residue. A careful examination of the NMR spectra of **2a** disclosed the presence of a minor component **2b,** which was subsequently proved to be an atropisomer of **2a**. Compound **2a** showed no optical activity, suggestive of it also being a racemate.

HR-ESIMS analysis revealed **2b** as an isomer of **2a.** By comparison with the ^1^H NMR spectra of **2a**, that of **2b** exhibited nearly identical signals to those of the minor component in **2a** (Appendix A). The methoxyl proton signals of **2b** (*δ* 3.97), downfield relative to those of **2a** (*δ* 3.04), inferred that the methoxy in **2b** was located in the deshielded area of the isoindole-1-one residue. **2a** and **2b** can also be interconverted with each other at room temperature.

As proved above, axial chirality was present for pestalachlorides A, A1, and A2, which resulted in time-dependent atropisomerism. To interrogate the stability of pestalachlorides atropisomers, we calculated the relative Gibbs energy barriers for the atropisomers interconversions at the M062X/def2TZVP/SMD (H_2_O)//B3LYP/6-31G(d)/PCM (H_2_O) level. The results disclosed that the barriers of **1a** to **1b** and **1b** to **1a** were 24.6 kcal/mol and 24.4 kcal/mol, and the corresponding interconversion half-times were 34 h and 24 h at room temperature, respectively, in agreement with the fact that **1b** is a little more unstable than **1a**; the barriers between two atropisomers of pestalachloride A were 21.4 kcal/mol and 21.6 kcal/mol, and the corresponding interconversion half-times were 0.15 h and 0.23 h, respectively, supporting their inseparability; the barriers of **2a** and **2b** interconversion were 26.9 kcal/mol and 27.4 kcal/mol, respectively, indicating that they can also interconvert with each other [29].

### 2.3. Proposed Biosynthetic Pathway for Pestalachlorides

Based on an analysis of the functions of genes within the *ptl* cluster, as well as a comparison with the previous reported biosynthetic assembly lines of geodin [26], pestheic acid [30], and monodictyphenone [31] (Table 2), a plausible biosynthesis pathway for pestalachlorides was proposed (Figure 1). The non-reducing polyketide synthase PtlB, showing 87%, 66%, and 63% of sequence identity with PtaB [30], mdpG [31], and GedC [26], respectively, was proposed to assemble and cyclize atrochrysone thioester (**5**). PltA, with high amino acid identity (92%) to PtaB [30], was reasoned to hydrolyze the thioester bond of **5** to release atrochrysone carboxylic acid (**6**) from PtlB. The following concerted decarboxylation and dehydration were proposed according to monodictyphenone biosynthetic logic. However, no gene encoding putative decarboxylase as MdpH was found within and near the *ptl* cluster, suggesting that the intermediate **6** might undergo spontaneous decarboxylation and dehydration to form emodin anthrone (**7**) as indicated by an earlier study [31,32]. **7** was subsequently oxidized to emodin (**8**) by a putative anthrone oxygenase PtlJ, which showed 44%, 41%, and 43% identity to GedH [26], PtaC [30], and MdpH2 [33], separately. According to the retrosynthesis analysis of pestalachlorides, **8** should be converted to alatinone (**9**); however, the mechanism of this conversion remains to be determined. Subsequently, as the previous study suggested [34], **9** might be cleaved to **10** via oxidation and thioesterification catalyzed by PtlC, a putative Baeyer-Villiger oxidase (47% identity with PtaJ [30]), and PtlF, a putative glutathione *S*-transferase (39% identity with MdpJ [31]), respectively. A succeeding reduction of **10** to the aldehyde **11** might be catalyzed by oxidoreductase PtlM, which displayed 51% identity to MdpK, and the latter was proposed to reduce thioester to benzaldehyde in arugosin F biosynthesis [34]. A putative prenyltransferase PtlH, the homolog of which is absent in Geodin, pestheic acid, and monodictyphenone biosynthetic assembly lines, showed 40% sequence identity with xanthone prenyltransferase A [35] and thus was postulated to C-prenylate **11** to give **4**. Subsequently, halogenation by FDH halogenase PtlK and methylation by O-methyltransferase PtlI occurred to give pestalone, which can be spontaneously reacted with primary amide to give compounds **1a/1b**, **2a/2b**, and **3** [36,37].

### 2.4. Antimicrobial Activities of Pestalachlorides

The analog of pestalachloride A1a and A2a, pestalachloride A, was previously reported to show antibacterial activity against the standard and methicillin-resistant Staphylococcus aureus (MIC = 10 *μ*g/mL) and the plant pathogenic fungus Fusarium culmorum (MIC = 3.2 *μ*g/mL). To preliminarily explore the bioactivity of new compounds and the structure-activity relationship of pestalachlorides, compounds A1a and A2a, together with pestalachloride A (3), were evaluated for their activity against standard *Staphylococcus aureus* ATCC 29213 and methicillin-resistant *Staphylococcus aureus* (MRSA), as well as other human pathogenic microbes, *Enterococcus faecium* ATCC 35667, Vancomycin-Resistant *Enterococcus faecium* (VRE), and *Candida albicans* ATCC 10231. The pestalachloride A showed moderate activity against *Staphylococcus aureus* ATCC 29213, MRSA, and VRE with minimum inhibitory concentrations (MIC) of 8 *μ*g/mL, 4 *μ*g/mL, and 8 *μ*g/mL, respectively. Pestalachloride A1a showed weak antibacterial activity against four Gram-positive bacteria (MIC = 32 *μ*g/mL), while pestalachloride A2a showed no antibacterial activity within the tested concentration range, indicating that the bulky *N*-substituents can reduce the antibacterial activity of these compounds (Table 3). On the other hand, all of the tested three compounds showed no activity against the fungus *Candida albicans*.

## 3. Discussion

With the development of sequencing and bioinformatics, genome mining has increasingly become an important strategy for identifying new compounds and cryptic enzymes and exploring new biosynthetic logics. Here we succeeded in discovering new pestalachloride analogs and thus unearthing their biosynthetic gene cluster by utilizing the strategy of halogenase-targeted genome mining combined with characteristic isotope patterns of halogen atoms. Pestalachlorides A1a, A2a, and their analog pestalachloride A share an isoinodin-1-one core structure that occurs in a number of bioactive compounds [37]. From the biosynthesis view, pestalachlides belong to the pestalone-type benzophenones [37]. This class of compounds features a prenyl group attached to a benzophone that is often clorinated. Although a total of 21 natural analogs of pestalone, including SB87-Cl and SB87-H from *Chrysosporium* sp. [38], pestalone from *Pestalotia* sp. CNL-365 [39], pestalachloride A-C from *Pestalotiopsis adusta* [27], (±)pestalachloride D from *Pestalotiopsis* sp. ZJ-2009-7-6 [40], pestalachlorides E and F from *Pestalotiopsis* sp. ZJ-2009-7-6 [41], pestalones B-H from *Pestalotiopsis neglecta* F9D003 [42], and pestalotinones A–D from *Pestalotiopsis trachicarpicola* SC-J551 [28] have been discovered, no biosynthesis gene clusters responsible for their assembly are reported. To our knowledge, this is the first report of the biosynthesis gene clusters of pestalachlorides and their analogs, pestalone-type benzophenones. So far, there are lots of known natural metabolites that are still not connected with their biosynthesis gene clusters, which hinders the further mining of natural products. Given a large part of them contain halogen atoms, halogenase-targeted genome mining reported here might be an efficient strategy to uncover their biosynthesis origin.

PtlK, assembling double chloride atoms to the phenol residue of pestalachlorides at the late stage of biosynthesis, was reasoned to be a free-standing phenol FDH. Free-standing FDHs, including indole and phenol FDHs, have gained broader interest because it is easier to use them in biotransformation. Amongst, free-standing indole FDHs have been deeply investigated and engineered [13,14,43,44,45,46]; however, the counterpart researches on free-standing phenol FDHs are still scarce. Although free-standing phenol FDHs are widely distributed in fungi, only a few are connected with their products, and none of their structures have been determined [7], which hinders the application of these enzymes. Further mining of fungal free-stand phenol FDHs and their products will benefit their structural determination and engineering for biocatalytic application.

## 4. Materials and Methods

### 4.1. General Experimental Details

UV measurements were recorded on a Shimadzu UV-2550 spectrophotometer. NMR spectra were acquired with Varian Mercury 600 spectrometers using DMSO-*d*_6_ as solvent. HR-ESIMS and ESIMS/MS data were obtained on a Waters Xevo G2-XS QTof mass spectrometer (Waters, Manchester, UK) with an ACQUITY UPLC^®^ CSH^TM^ C_18_ column (Waters, 1.7 *μ*m, 2.1 × 100 mm) or CORTECS^®^ C_18_ (waters, 2.7 *μ*m, 2.1 × 50 mm) HPLC analyses were performed on an Agilent 1200 or Shimadzu LC-20A instrument using an XBridge C_18_ column (3.5 *μ*m, 4.6 × 150 mm) or Reprosil-Pur Basic-C_18_ column (5 *μ*m, 250 × 10 mm). The genomic DNA was sequenced using the IlluminaHiSeq platform (Illumina, San Diego, CA, USA), assembled via SPAdes 3.13.0 software [47], and uploaded onto Genbank (JALYBT000000000).

### 4.2. Genome Mining of the Halogenase-Containing Biosynthesis Gene Clusters

TBlastp analysis was performed using fungal FDH GedL as the probe to explore new halogenated secondary metabolites from our in-house fungal genomic database. The hit-containing sequences were further analyzed by antiSMASH and the putative halogenase potentially involved in secondary metabolite biosynthesis were picked out for further phylogenetic analysis with characterized FDHs. The characterized FDHs are Rdc2 (ADM86580), KtzR (ABV56598), RebH (AAN01216), ThaL (ABK79936), PrnA (AAB97504), BhaA (CAA76550), SttH (ADW94630), MibH (ADK32563), TiaM (ADU85999), PyrH (AAU95674), VhaA (CCD33142), CndH (CAQ43074), CrpH (ABM21576), SgcC3 (ANY94426), GsfI (ADI24948), AcOTAhal (ANY27070), AclH (BAE56588), KtzQ (ABV56597), PtaM (AGO59046), CalO3 (AAM70353), ChlB4 (AAZ77674), ChmK (BBL33413), ChmN (BBL33407), PloN (BBL33421), PloK (BBL33420), AscD (A0A455R7M0), Ota5 (A2R6G7), GedL (XP_001217599), RadH (ACM42402), ArmH5 (ALT31852), ArmH4 (ALT31851), ArmH1 (AEM76785), ArmH2 (AEM76786), ArmH3 (ALT31850), ChlA (BAP16678), and VirX1 (6QGM). The amino acid sequence of the putative halogenase PtlK combined with the selected known halogenases was aligned by MUSCLE [48], and their phylogenetic tree was constructed based on the UPGMA [49] method and visualized with MEGA 7.0.26 [50].

### 4.3. Culture Condition Prioritization for the Production of Chlorinated Compounds

*Pestalotiopsis rhododendri* LF-19-12 was originally isolated from a lichen sample collected from Tibet, China, and identified based on phylogenetic NJ tree based on ITS sequences (Appendix A). To explore the production of chlorinated compounds, four culture media, M1 (peptone 2 g, yeast powder 4 g, starch 10 g, 1 L distilled water), M2 (mannitol 40 g, maltose 40 g, yeast powder 10 g, K_2_HPO_4_ 2 g, MgSO_4_·7H_2_O 0.5 g, FeSO_4_·7H_2_O 0.01 g, 1 L distilled water), PDB (200 g potato, 20 g glucose, 1 L distilled water), and YES media (sucrose 150 g, yeast powder 20 g, MgSO_4_·7H_2_O 0.5 g, ZnSO_4_·7H_2_O 0.01 g, CuSO_4_·5H_2_O 0.005 g, 1 L distilled water) were selected for culturing *Pestalotiopsis rhododendri* LF-19-12. The fungus *Pestalotiopsis* LF-19-12 was first cultured in 250 mL Erlenmeyer flasks containing 50 mL of potato dextrose broth (PDB) medium and incubated on a rotary shaker at 220 rpm and 28 °C for 48 h to yield the seed culture. Then 50 mL of the seed culture was inoculated into a 500 mL Erlenmeyer flask containing 100 mL of fermentation medium and incubated at 220 rpm and 28 °C for 9 days. The fermentation for each culture medium was carried out in triplicate. Subsequently, 2 mL of culture was filtrated, and the obtained mycelia were extracted using methanol. The obtained crude extract was pretreated with ODS and then analyzed using HR-ESIMS/MS.

### 4.4. Fermentation and Isolation

The spores of *Pestalotiopsis* LF-19-12 were inoculated into 3 × 500 mL Erlenmeyer flasks each containing 100 mL of potato dextrose broth (PDB) medium to be precultured at 28 °C and 220 rpm for 48 h. Then, the obtained 3 × 100 mL of preculture were inoculated into 3 × 5 L Erlenmeyer flask each containing 1 L of M2 medium (mannitol 40 g, maltose 40 g, yeast powder 10 g, K_2_HPO_4_ 2 g, MgSO_4_·7H_2_O 0.5 g, FeSO_4_·7H_2_O 0.01 g, 1 L distilled water), and incubated on a rotary shaker at 220 rpm and 28 °C. After 9 days, the mycelia were harvested and extracted six times with acetone, yielding 17.72 g of crude extract. The obtained extract was subjected to a silica gel (300–400 mesh, Yantai Chemical Industry Research Institute, Yantai, China) column eluted with a stepwise gradient of CH_2_Cl_2_−MeOH mixtures (1:0, 100:1, 100:2, 100:4, 100:6, 100:8, 10:1, 5:1, 4:1, 2:1, 0:1, *v*/*v*) to give Fr A-G. The eluents were analyzed by LC-MS, and the targeted compounds were mainly found in the Fr.E (576.95 mg) and Fr.F (172.64 mg).

Fr.E was further separated with an ODS flash column eluted with a gradient ACN−H_2_O solution from 15% ACN to 70% ACN. **1a** (2.3 mg) was crystalized from the eluent of 45% ACN. The other eluents were combined into five fractions (E1: 72.5 mg; E2: 52.3 mg; E3: 57.1 mg; E4: 36.7 mg; E5: 145.7 mg) according to LC-MS analysis results. The Fr. E3 was further purified by semi-preparative RP HPLC (Reprosil-Pur Basic-C_18_, 5 *μ*m, 250 × 10 mm, 30% ACN−H_2_O, 30 °C, 2.5 mL/min) to yield **1a**
*(t_R_* 40.54 min, 20.9 mg) and **1b** (*t*_R_ 54.50 min, 2.2 mg). The Fr. E4 was further purified by semi-preparative RP HPLC (Xbridge™ Prep C_18_, 5 *μ*m, 250 × 10 mm, 40% ACN−H_2_O, 30 °C, 2.5 mL/min) to afford **3** (*t_R_* 48.03 min, 2.6 mg). The Fr. E5 was purified using semi-preparative RP HPLC (Xbridge™ Prep C_18_, 5 *μ*m, 250 × 10 mm, 40% ACN−H_2_O, 30 °C, 2.5 mL/min) to give **4** (*t*_R_ 17.25 min, 13.5 mg).

Fr.F was separated with an ODS flash column eluted with a stepwise gradient of ACN in water (20%, 30%, 50%, 70%, and 100%; *v*/*v*; each for 5 min). All eluents were analyzed by LC−MS, and those containing halogenated compounds were combined to yield fraction Fr.F1 (37.3 mg). The Fr.F1 was subsequently purified by semi-preparative chromatography (XSelect CSH C_18_ OBD ^TM^ prep column, 5 *μ*m, 250 × 10 mm, 45% ACN aqueous containing 0.1% TFA, 30 °C, 2.5 mL/min), to yield **A2a** (*t*_R_ 30.5 min, 1.9 mg) and **A2b** (*t*_R_ 48.8 min,0.2 mg).

Pestalachloride A1a (**1a**): white powder; 0 (*c* 0.1, MeOH); UV (MeOH) *λ*_max_ (log *ε*) 222.4 (4.78) 258.8 (32.43), 297.2 (26.26); 1D and 2D NMR data (DMSO-*d*_6_) see Table 1 and Appendix A; HR-ESI(+)MS [M+H]^+^ *m/z* 482.1121 (calcd. for C_23_H_26_Cl_2_NO_6_, 482.1137, 3.3 ppm).

Pestalachloride A1b (**1b**): white powder; ^1^H and ^13^ C NMR data (DMSO-*d*_6_) see Table 1 and Appendix A; HR-ESI(+)MS [M+H]^+^
*m/z* 482.1117 (calcd. for C_23_H_26_Cl_2_NO_6_, 482.1137, 4.1 ppm).

Pestalachloride A2a (**2a**): white powder; 0 (*c* 0.1, MeOH); UV (MeOH) *λ*_max_ (log *ε*) 208.2 (3.6), 258.8 (38.7), 297.0 (30.4); 1D and 2D NMR data (DMSO-*d*_6_) see Table 1 and Appendix A; HR-ESI(+)MS [M+H]^+^ *m/z* 524.1240 (calcd. for C_25_H_28_Cl_2_NO_7_, 524.1243, 0.57 ppm).

Pestalachloride A2b (**2b**): white powder; ^1^H NMR data (600 MHz, DMSO-*d*_6_) see Table 1 and Appendix A; HR-ESI(+)MS [M+H]^+^ *m/z* 524.1237 (calcd. for C_25_H_28_Cl_2_NO_7_, 524.1243, 1.1 ppm).

### 4.5. The Calculation of the Relative Gibbs Energy Barriers

*S*-configuration structures of **1a**, **1b**, **2a**, **2b**, and atropisomers of **3** were first optimized using Gaussian 16 at the B3LYP/6-31G (d)/PCM (H_2_O) level. Then relaxed dihedral angle (rotation between C8–C9) scans were performed at the same level. The Gibbs energies for the calculation of barriers were calculated at the M062X/def2TZVP/SMD (H_2_O)//B3LYP/6-31G (d)/PCM (H_2_O) level.

### 4.6. Antibacterial Bioassay

The minimum inhibitory concentration (MIC) values of the obtained compounds against *Staphylococcus aureus* ATCC 29213, *Enterococcus faecium* ATCC 35667, Methicillin-Resistant *Staphylococcus aureus*, Vancomycin-Resistant *Enterococcus faecium*, and *Candida albicans* ATCC 10231 were determined using a broth microdilution protocol [51]. Briefly, 50 *μ*L of bacterial or fungal suspension (5 × 10^5^ CFU/mL) was added to each well of the 96-well plate. Subsequently, 50 *μ*L of each work solution of pestalachlorides A1a, A2a, A and the corresponding positive drugs (64, 32, 16, 8, 4, 2, 1, 0.5, 0.25, 0.125, 0.0625, 0.03125 *μ*g/mL) were added and incubated at 33 °C for 18 h. The lowest concentration that completely prevents the growth of the assayed organism was defined as the MIC.

## Data Availability

The data presented in this study are available on request from the corresponding author.

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
