# Peer review of "Halogenase-Targeted Genome Mining Leads to the Discovery of (±) Pestalachlorides A1a, A2a, and Their Atropisomers"

_antibiotics, 2022, doi:10.3390/antibiotics11101304_

Round 1
Reviewer 1 Report
Genome mining has become an important tool for discovering new natural products and identifying the cryptic biosynthesis gene clusters. Mengna Luo et al. has investigated Halogenase-Targeted Genome Mining Leads to the Discovery of (±) Pestalachlorides A1a, A2a, and Their Atropisomers.
After an exhaustive revision, the manuscript is major revision (article has serious flaws). In general, the study is closely connected to the journal's objectives. The study is very interesting.
I give a detailed revision of the manuscript:
1. The introduction is too concise, and it is very, very short. The authors need to rewrite the introductions, with precise information.
2. The section result, these subsections are very poor. The section discussion is an important problem, since is very poor.
3. Literature:
The references are wrong, the authors need to see the guide for authors. The literature should be improved, including the following items:
1. Parisini, E.; Metrangolo, P.; Pilati, T.; Resnati, G.; Terraneo, G. Halogen bonding in halocarbon-protein complexes: a 315 structural survey. Chem Soc Rev 2011, 40, 2267-2278, doi:10.1039/c0cs00177e.
2. Crowe, C.; Molyneux, S.; Sharma, S.V.; Zhang, Y.; Gkotsi, D.S.; Connaris, H.; Goss, R.J.M. Halogenases: a palette of emerging 317 opportunities for synthetic biology–synthetic chemistry and C–H functionalisation. Chemical Society Reviews 2021, 50, 9443- 318 9481, doi:10.1039/D0CS01551B.
22. Li, E.; Jiang, L.; Guo, L.; Zhang, H.; Che, Y. Pestalachlorides A–C, antifungal metabolites from the plant endophytic fungus 367 Pestalotiopsis adusta. Bioorganic & Medicinal Chemistry 2008, 16, 7894-7899, doi:10.1016/j.bmc.2008.07.075.
23. g, Z.; Wu, P.; Li, H.; Xue, J.; Wei, X. Pestalotinones A-D, new benzophenone antibiotics from endophytic fungus 369 Pestalotiopsis trachicarpicola SC-J551. J Antibiot (Tokyo) 2022, 75, 207-212, doi:10.1038/s41429-022-00510-0.
25. Xu, X.; Liu, L.; Zhang, F.; Wang, W.; Li, J.; Guo, L.; Che, Y.; Liu, G. Identification of the first diphenyl ether gene cluster for 373 pestheic acid biosynthesis in plant endophyte Pestalotiopsis fici. Chembiochem 2014, 15, 284-292, doi:10.1002/cbic.201300626.
and others.
3. Discussion:
The section of “Discussion” is not characterized by an explication of the results, comparison with other studies, and explication (discussion) of the results obtained with respect to other studies.
4. The manuscript has no information about:
Figure S1. 1H NMR for pestalachloride A1a (1a) in DMSO-d6 (600 MHz)
Figure S2. 13C NMR for pestalachloride A1a (1a) in DMSO-d6 (150 MHz)
Figure S3. gCOSY for pestalachloride A1a (1a) in DMSO-d6
Figure S4. HSQC for pestalachloride A1a (1a) in DMSO-d6
Figure S5. HMBC for pestalachloride A1a (1a) in DMSO-d6
Figure S6. 1H NMR for pestalachloride A1b (1b) in DMSO-d6 (600 MHz)
Figure S7. 13C NMR for pestalachloride A1b (1b) in DMSO-d6 (150 MHz)
Figure S10. HSQC for pestalachloride A2a (2a) in DMSO-d6
5. Provide the full name of Pestalotiopsis sp.
Author Response
Dear editor:
Thank you and three reviewers so much for your professional suggestions. We have revised the manuscript according to the suggestions one by one.
First, responses to the questions of reviewer 1 are as the following:
- The introduction is too concise, and it is very, very short. The authors need to rewrite the introductions, with precise information.
Response: The part of the introduction has been rewritten to give more background.
- The section result, these subsections are very poor. The section discussion is an important problem, since is very poor.
Response: the subsections of results have been rearranged in new manuscript, and the section of the discussion has been revised to give more analysis of the results, especially compared with related reports.
- Literature:
The references are wrong, the authors need to see the guide for authors. The literature should be improved, including the following items:
- Parisini, E.; Metrangolo, P.; Pilati, T.; Resnati, G.; Terraneo, G. Halogen bonding in halocarbon-protein complexes: a 315 structural survey. Chem Soc Rev2011, 40, 2267-2278, doi:10.1039/c0cs00177e.
- Crowe, C.; Molyneux, S.; Sharma, S.V.; Zhang, Y.; Gkotsi, D.S.; Connaris, H.; Goss, R.J.M. Halogenases: a palette of emerging 317 opportunities for synthetic biology–synthetic chemistry and C–H functionalisation. Chemical Society Reviews 2021, 50, 9443- 318 9481, doi:10.1039/D0CS01551B.
- Li, E.; Jiang, L.; Guo, L.; Zhang, H.; Che, Y. Pestalachlorides A–C, antifungal metabolites from the plant endophytic fungus 367 Pestalotiopsis adusta. Bioorganic & Medicinal Chemistry 2008, 16, 7894-7899, doi:10.1016/j.bmc.2008.07.075.
- g, Z.; Wu, P.; Li, H.; Xue, J.; Wei, X. Pestalotinones A-D, new benzophenone antibiotics from endophytic fungus 369 Pestalotiopsis trachicarpicola SC-J551. J Antibiot (Tokyo)2022, 75, 207-212, doi:10.1038/s41429-022-00510-0.
- Xu, X.; Liu, L.; Zhang, F.; Wang, W.; Li, J.; Guo, L.; Che, Y.; Liu, G. Identification of the first diphenyl ether gene cluster for 373 pestheic acid biosynthesis in plant endophyte Pestalotiopsis fici. Chembiochem 2014, 15, 284-292, doi:10.1002/cbic.201300626.
and others.
Response: Some erroneous references have been revised.
- Discussion:
The section of “Discussion” is not characterized by an explication of the results, comparison with other studies, and explication (discussion) of the results obtained with respect to other studies.
Response: the section of the discussion has been revised to give more analysis of the results, especially compared with other related studies.
- The manuscript has no information about:
Figure S1. 1H NMR for pestalachloride A1a (1a) in DMSO-d6 (600 MHz)
Figure S2. 13C NMR for pestalachloride A1a (1a) in DMSO-d6 (150 MHz)
Figure S3. gCOSY for pestalachloride A1a (1a) in DMSO-d6
Figure S4. HSQC for pestalachloride A1a (1a) in DMSO-d6
Figure S5. HMBC for pestalachloride A1a (1a) in DMSO-d6
Figure S6. 1H NMR for pestalachloride A1b (1b) in DMSO-d6 (600 MHz)
Figure S7. 13C NMR for pestalachloride A1b (1b) in DMSO-d6 (150 MHz)
Figure S10. HSQC for pestalachloride A2a (2a) in DMSO-d6
Response: The corresponding information has been added in the revised manuscript.
- Provide the full name of Pestalotiopsis sp.
Response: Full name Pestalotiopsis rhododendri LF-19-12 has substituted the original name.

Reviewer 2 Report
This article describes Genome mining studies conducted for halogenated Pestalachlorides and their characterization. Although the work theme seems unique there are several issues with the present version as appended below:
- Why antimicrobial/MIC study was conducted? the MIC results also did not reflect any significant compound. The relevance of this aspect within the manuscript is unclear.
- The materials and methods are not clearly described. How did the author learn about the solvents used and the extraction methodology adopted for this purpose? No reference was cited for this in sections 4.3 and 4.4. How did the author rule out the chances of sample oxidation?
- The conditions used for UPGMA and other tools used during the study must be clearly described, or appropriate references must be provided.
- How the authors got the fungal culture?
- Full form of BGC?
- No discussion of obtained results; only future prospects were provided in this section.
- Line 67, the function of ptaK was not described!
- The term 'afford' is unusual but appears several times in the MS.
Author Response
Dear reviewer:
Thanks so much for your professional suggestions. We have revised the manuscript according to the suggestions one by one.
- Why antimicrobial/MIC study was conducted? the MIC results also did not reflect any significant compound. The relevance of this aspect within the manuscript is unclear.
Response: The analogue of pestalachloride A1a and A2a, pestalachloride A, was previously reported to show antibacterial activity against standard and methicillin-resistant Staphylococcus aureus (MIC = 10 μg/mL) and plant pathogenic fungus Fusarium culmorum (MIC = 3.2 μg/mL). To preliminarily explore the bioactivity of new compounds and structure-activity relationship of pestalachlorides, the antimicrobial study was conducted. The corresponding expression has been added in the manuscript. Although no significant activity was obtained, we still think this study is valuable for new compounds.
- The materials and methods are not clearly described. How did the author learn about the solvents used and the extraction methodology adopted for this purpose? No reference was cited for this in sections 4.3 and 4.4. How did the author rule out the chances of sample oxidation?
Response: The solvents and the extraction method are routinely used in our lab and they are suitable for nonpolar natural products. Halogenated compounds often showed nonpolarity due to the presence of halogen. This is the reason why we chose this method. And, fortunately, this method worked well for our purpose. As for the question about sample oxidation, I am sorry that no chances of sample oxidation found in our experiments, but atropisomerism are present for compound 1a/1b and 2a/2b. At first, we did not know they are atropisomers and we just known they can be interconverted according to the HPLC-MS analysis. After we elucidated the planar structures, we found their analogue pestalachloride A is a mixture of atropisomers due to the rotation hindrance of C8-C9. The rotation hindrance can be enforced by the bulky substituents in the aryl ring which further led us to rule out their chances.
- The conditions used for UPGMA and other tools used during the study must be clearly described, or appropriate references must be provided.
Response: Relevant references has been added.
- How the authors got the fungal culture?
Response: Fungus Pestalotiopsis sp. LF-19-12 was kept in our in-house fungal culture collection and it was originally isolated from a lichen sample collected from Tibet, China. The genome and ITS sequences together with the detail information on the origin of pestalotiopsis sp. LF-19-12 have been submitted to NCBI (accessions: JALYBT000000000 and OM807202.1) and added in the revised manuscript
- Full form of BGC?
Response: We have released the genome sequence of pestalotiopsis sp. LF-19-12 (accessions: JALYBT000000000). Contig 94 includes this BGC. Now we resubmitted the complete BGC and the accession no is OP253974 which is being processed and then will be released.
- No discussion of obtained results; only future prospects were provided in this section.
Response: The discussion about the obtained results has been added.
- Line 67, the function of ptaK was not described!
Response: the function of ptaK has been added.
The sentence “… …a gene ptaK, encoding a putative protein with 51% amino acid… …” was changed to “… …a gene ptaK, encoding a putative flavin-dependent halogenase with 51% amino acid… …”
- The term 'afford' is unusual but appears several times in the MS.
Response: “afford” has been substituted by its synonyms “yield” and “give”

Reviewer 3 Report
check the drawings of the chemical structures of figure 1 (structures 1-2)
explains figure 3. there is not much correspondence between the figure and the comment, what does figure b represent? a mass specter?
better describe the conditions of the chromatographic separation and indicate the percentage yield with respect to the purified crude
write element numbers as a subscript, not as a shortened character
describe better the chemical part par. 2.3. perhaps it should be simplified and made more understandable;
review table 1;
simplifies Figure 4
check scheme 1. it is not very clear (names and acronyms of the molecules)
enter a part in which you explain what acronyms are used in the text. they are not always clear
Author Response
Dear reviewer:
Thanks so much for your professional suggestions. We have revised the manuscript according to the suggestions one by one.
- explains figure 3. there is not much correspondence between the figure and the comment, what does figure b represent? a mass specter?
Response: Figure b represent the mass spectra of the major peaks, which has been added in the manuscript.
- better describe the conditions of the chromatographic separation and indicate the percentage yield with respect to the purified crude
Response: the progress of chromatographic separation has been rewritten and the yields have been added
- write element numbers as a subscript, not as a shortened character
Response: element numbers have been revised to substripts.
- describe better the chemical part par. 2.3. perhaps it should be simplified and made more understandable;
Response: The structural elucidation part has been simplified in the new version of manuscript.
- review table 1;
Response: We have reviewed table 1 and corrected some mistakes in the new version of manuscript.
- simplifies Figure 4
Response: Figure 4 has been simplified in the new version of manuscript.
- check scheme 1. it is not very clear (names and acronyms of the molecules). enter a part in which you explain what acronyms are used in the text. they are not always clear
Response: We would think using acronyms for the intermediates could make the description of biosynthesis more readable. We are sorry for this unclearity. We referred other papers and numbered the intermediates again using continuous numbers in this manuscript.
Round 2
Reviewer 1 Report
Accepted in current form
Author Response
Dear reviewer:
Thanks so much for your professional suggestions.
Reviewer 2 Report
reject
Author Response

(The authors gave the same response as above.)

Reviewer 3 Report
In the chemical part, in addition to the quantities reported in mg, it would also be appropriate to indicate the moles, to make the expression of the yield more understandable;
line 298-300: better express the percentage of solvents used in the elution: it is not clear;
line 303: write ACN instead of acetonitrile (previously you indicated the solvent abbreviation);
lines 316-328. check the data presented: there are typing errors in the alpha D or ESI-MS writing; also the ways of indicating the brute formulas of the molecules have not been corrected (the numbers should be written as subscripts and not with the reduction of the character)
several typos (missing spaces between words);
Author Response
Dear reviewer:
Thanks so much for your professional suggestions. We have revised the manuscript according to the suggestions one by one.
- In the chemical part, in addition to the quantities reported in mg, it would also be appropriate to indicate the moles, to make the expression of the yield more understandable;
Response: Thank you for your valuable comments. Unfortunately the crude extract are mostly mixtures and cannot be expressed in moles, so we could only calculated the yield based on the weight of the products.
- line 298-300: better express the percentage of solvents used in the elution: it is not clear;
Response: The express of the percentage of solvents has been changed from “ 1 L CH2Cl2→1 L CH2Cl2:MeOH (100:1) →1 L ... ... ” to “1:0, 100:1, 100:2,100:4, 100:6, 100:8, 10:1, 5:1, 4:1, 2:1, 0:1,v/v”
- line 303: write ACN instead of acetonitrile (previously you indicated the solvent abbreviation);
Response: All the “acetonitrile” have been changed to “ACN”
- lines 316-328. check the data presented: there are typing errors in the alpha D or ESI-MS writing; also the ways of indicating the brute formulas of the molecules have not been corrected (the numbers should be written as subscripts and not with the reduction of the character)
Response: The alpha D has been corrected and ESIMS in lines 316-328 are uniformly written as ESI-MS.
However I'm sorry we can't find anyone that the numbers of formulas of the molecules written as the reduction of the character. It may be that the font makes the subscript look like the reduction of the character.
- several typos (missing spaces between words);
Response: We have checked the full text and corrected them.
